# Rosmarinic Acid Production from *Origanum dictamnus* L. Root Liquid Cultures In Vitro

**DOI:** 10.3390/plants12020299

**Published:** 2023-01-08

**Authors:** Virginia Sarropoulou, Charikleia Paloukopoulou, Anastasia Karioti, Eleni Maloupa, Katerina Grigoriadou

**Affiliations:** 1Institute of Plant Breeding and Genetic Resources, Hellenic Agricultural Organization ELGO-DIMITRA, 57001 Thessaloniki, Greece; 2Laboratory of Pharmacognosy, School of Pharmacy, Aristotle University of Thessaloniki, University Campus, 54124 Thessaloniki, Greece

**Keywords:** adventitious roots, *Origanum dictamnus* L., Cretan dittany, HPLC-PDA-MS, rosmarinic acid, secondary metabolites, plant growth regulators

## Abstract

In the present work *Origanum dictamnus* L. was studied as a suitable in vitro adventitious root culture system for the production of important bioactive molecules, such as rosmarinic acid (RA). Callus culture was initiated from leaf, petiole and root explants on solid MS medium supplemented with either 5 μM NAA + 5 μM kinetin (ODK3) or 5 μM NAA + 0.5 μM kinetin (ODK4). New roots formed from leaf, petiole and root calluses were aseptically transferred into Erlenmeyer flasks containing 100 mL liquid medium and shaken at 120 rpm in the dark. The liquid medium used was the MS supplemented either with 35 μM IBA + 2.5 μM kinetin (ODY1) or 5 μM NAA + 0.5 μM kinetin (ODY2). Biomass production parameters, RA content (%) and yield index (YI) were recorded for each treatment explant type, medium composition and incubation period. Results showed, in every case, the production of RA in vitro. Between the two liquid media (ODY1, ODY2) and the different culture periods, the ODY1 medium and the longest 200-day-culture period were more effective for RA and biomass production, regardless of the initial explant type used. The combination of ODK4-ODY1 resulted in higher RA (5.1% and 4.7%), fresh biomass production (19.0 g and 11.6 g), mean YI (93.7 mg and 51.4 mg) and YI per explant (3.75 mg and 2.06 mg) for roots derived from leaf calluses and root calluses, respectively. However, the solid ODK3 (200 days)–liquid ODY1 (40 days) transition treatment was more beneficial for roots derived from petiole calluses leading to an 18.8-fold increase in fresh biomass growth rate. RA accumulation and YIs were also significantly influenced by explant type, with the highest value produced from root petiole calluses (6.6% RA dry weight, 115.3 mg mean YI and 4.61 mg YI per explant) after 240 days.

## 1. Introduction

Rosmarinic acid (RA) (Figure 1) is a phenolic secondary metabolite widely found in medicinal and aromatic plants of the Lamiaceae family. Although its name originates from rosemary (*Rosmarinus officinalis*), it is biosynthesized in considerable amounts in *Thymus* sp., *Origanum* sp., *Melissa* sp., *Salvia* sp., *Satureja* sp., *Mentha* sp. and other species [1]. Chemically is the simplest member of the group of depsides, being an ester of caffeic acid and 3,4-dihydroxyphenyllactic acid [2]. Due to the presence of two electroactive catechol moieties has excellent antioxidant properties [3]; while it possesses metal-chelating properties, which further contribute to the antioxidant potential of the molecule, its antioxidant activity has been reported to be even stronger than that of vitamin E [4]. These antioxidant activities have been established in several in vitro and in vivo studies, and their beneficial role in conditions where oxidative stress has a key role, like inflammation, diabetes, neuroprotection and hepatoprotection, has been supported by many researchers. Clinical studies with plant extracts rich in RA have supported its role as an adjuvant to the conventional drugs in the treatment of several conditions, such as osteoarthritis, as a nootropic agent, in allergic rhinoconjunctivitis and in metabolic syndrome [3,5,6], thereby reducing the risk of cancer, atherosclerosis and other diseases associated with augmented oxidative stress [4]. Due to the wide range of pharmacological attributes, RA is considered a commercially important product for the pharmaceutical and cosmetic industries [7,8]. Its applications extend further to the cosmetic–dermocosmetic industry and in the food industry as an alternative/additional antioxidant agent for food preservation [9]. Therefore, the production of RA alone or in enriched plant extracts is desirable.

*Origanum dictamnus* L., or Cretan dittany, is a perennial and aromatic plant that only grows wild in the mountainsides and gorges of the island of Crete in Greece [10]. The extracts of *O. dictamnus* have been reported to exert several biological activities, including antioxidant [11,12,13,14], antimicrobial against bacteria, protozoans and fungi [15,16,17,18,19], cytotoxic activity against P388 (murineleukemia) and human bronchial epidermoid lung cancer NSCLC-N6 cell lines, respectively [20]. From the polar extracts of the aerial parts of *O. dictamnus*, monoterpenes, alicyclic derivatives, flavonoids and depsides (salvianolic acid P, rosmarinic acid and rosmarinic acid methyl ester) have been isolated and structurally determined [19]. Recently, a monograph for Dittany of Crete was approved among EU countries registering it as a traditional herbal medicinal product, i.e., herbal tea for oral and cutaneous use (as infusion or decoction) [21,22].

In contrast to the majority of other secondary metabolites (SMs), RA is actively synthesized during tissue growth [23,24]. RA is accumulated very intensively in non-differentiated cell cultures, often in much larger quantities than in intact plants [25]. Numerous species have been used to produce RA in plant in vitro cultures of Lamiaceae species, such as in callus and cell suspension cultures of *Ocimum sanctum* [26], *Coleus blumei* [27], *Salvia milttiorhiza* [28], *Salvia officinalis* L. [29] and *Salvia fruticosa* [30], cell suspension cultures of *Lavandula vera* MM. [31] and root cultures of *Ocimum basilicum* L. [32]. 

Adventitious root cultures show a higher constancy in the production of highly active compounds with more rapid growth than that of cell cultures [33]. In addition, bioreactor system cultivation for plant bioactive compounds using adventitious root cultures has emerged as a technology with possible commercial applications [34]. Plant roots are the main raw materials that contribute to herbal drug preparations, accounting for about 60% of herbal medicinal plants applied for ethno-medicine needs; therefore, adventitious root culture establishment is highly useful [35]. Although members of the Nepetoideae subfamily of Lamiaceae family represent a good source of RA [1,5], the amount of RA depends on various factors, such as genetic factors, environmental factors and the time of harvest. 

The aim of the study was the establishment of a suitable in vitro adventitious root culture system for the production of RA from the Greek endemic *Origanum dictamnus*.

## 2. Results

### 2.1. Evaluation of Rosmarinic Acid Production by HPLC-UV and NMR

All root extracts derived from leaf, petiole and root callus cultures were analyzed by HPLC-PDA-MS and showed to contain mainly rosmarinic acid (Appendix A). Long HPLC elution program confirmed the absence of similar RA metabolites, such as caffeic acid, 3,4-dihydroxyphenylactic acid, salvianolic acid P and its derivatives. These compounds were observed in *O. dictamnus* leaves and petioles (Appendix A). Lipophilic components, such as fatty acids, were also present, as revealed by examination of the extracts by NMR (Appendix A). In brief, two ABX systems were observed in the aromatic area of the ^1^H-NMR spectrum corresponding to the subgroups of caffeic and 3,4-dihydroxyphenylactic acid. In the same spectrum, two doublets of ~16.0 Hz were observed, typical of the two *E* protons of the caffeoyl moiety. The area between 0.0 and 3.0 ppm exhibited signals belonging to aliphatic protons, probably lipophilic constituents. Typical of fatty acid is the strong peak at approximately 1.22 ppm. For a more detailed NMR analysis, see Appendix A. Quantitative analysis of rosmarinic acid was performed by HPLC-UV, and results are shown in Table 1, Table 2, Table 3 and Table 4.

### 2.2. In vitro Adventitious Root Culture Establishment

After two months of culture, the formation of callus and root regeneration at a rate of 100% were obtained regardless of the explant type (leaf, petiole, root) and the composition of the solid culture medium (ODK3: 5 μM KIN + 5 μM NAA, ODK4: 0.5 μM KIN + 5 μM NAA). Shoot regeneration did not occur in any treatment (Figure 2a–f). Different culture periods are needed for the differentiation of the three different explant types (leaves, petioles, roots) to callus and subsequently for root regeneration via callus. In particular, the formation of callus in leaf explants started earlier after a culture period of 10 days, as compared to petiole explants which started to differentiate into callus after 20 days of culture, while petioles needed a longer culture period; at least 40 days for initiation of callus formation, either in ODK3 or ODK4 agar solidified media. After callus formation for all three explant types, a further culture period was essential for the initiation of root regeneration. In specific, the essential culture period for callus and subsequent root induction was 25 days for leaves, 35 days for roots and 90 days for petioles. In all three explant types, newly formed adventitious roots had to be enough developed in length so as their dissection from callus to be easy for further liquid cultures. Depending on the type of explant and the composition of the culture medium, there were differences regarding the quantities in which calli were formed and the morphology of the induced adventitious roots. For instance, the new roots regenerated via leaf callus were white colored, non-fluffy, longer and thinner, and the formed callus was smaller in size and limited in extent only to the abaxial surface of the leaf and to the cut/injury point of the leaf along the petiole as compared to callus and new induced roots from petiole and root explants. In the case of petioles and roots, the formed callus was lighter brown in color and larger in size, covering the entire surface of the original explant, and the induced roots were yellow light brown in color, thicker and shorter in length with the presence of fluff. For leaf and root explants, calli formed and thereafter, roots were induced in greater quantities on the ODK4 culture medium, whereas callus and root regeneration via petioles were more profound in the case of the ODK3 culture medium. Therefore, larger amounts of calli were produced in the descending order of the initial explant, petiole > root > leaf, while a larger amount of adventitious roots were induced in the descending order of the callus explant; leaf callus > root callus > petiole callus (Figure 2a–f).

### 2.3. In vitro Adventitious Root Culture

#### 2.3.1. Adventitious Roots from Leaf Callus

Adventitious roots on liquid ODY1 (40 days) previously cultured on ODK4 agar solidified medium (200 days) gained the highest biomass values (19 g FW and 1.8 g DW), while the highest accumulation of RA (5.1–5.8%) was recorded after 240 days of total culture at ODK3–ODY1, ODK4–ODY1 combinations and ODK3–ODY2 (120–40 days). The mean yield index (YI) of RA and yield index (YI) per explant reached their highest values (86.7–93.7 mean and 3.47–3.75/explant) when roots were incubated at the first two combinations for a total of 240 days. The general linear model and three-way ANOVA revealed that the composition of the initial agar solidified medium tested did not play a significant role (*p*-values > 0.05), while liquid media composition and duration of culture significantly affected biomass and RA production (Table 1).

#### 2.3.2. Adventitious Roots from Petiole Callus

Biomass (18.8 g FW, 1.8 g DW), RA production (6.6%) and YI (115.3 mean and 4.61 per explant) were significantly higher when roots cultured in ODK3–ODY1 combination for a total of 240 days and gave the best performance between the factors and interactions tested. The composition of the initial agar solidified medium was found to be non-significant (Table 2). In the case of roots derived from petiole callus, only two incubation periods in initial agar solidified media (120 and 200 days) were tested because a longer cultivation period is needed for the differentiation of petioles to callus and subsequently for root regeneration via callus.

#### 2.3.3. Adventitious Roots from Root Callus

Fresh (16.2 g) and dry weights (1.7 g) were significantly higher after 240 days of total culture in the ODK3–ODY1 combination, whereas the accumulation of RA reached its highest value (4.7%) in ODK4–ODY1 combination. Mean YI and YI per explant reached their highest values (47.4–51.4 mean and 1.90–2.06/explant) at both ODK3–ODY1 or ODK4–ODY1 combinations. Similar to the results of the other two types of explants, the composition of the initial agar solidified medium was found to be non-significant (Table 3).

#### 2.3.4. Combined Effect of Explant Type (Leaf, Petiole and Root Callus Derived Roots) and Medium Composition under 240 Days of Total Incubation

Based on the results described above, all three root explant types used responded better in the longest total culture period of 240 days and in the ODY1 liquid medium, irrespective of the initial agar solidified medium. The results showed that fresh biomass (18.8 g), RA production (6.6%) and YI (115.3 mean and 4.61 per explant) were significantly higher at the same time for petiole callus origin roots cultured in ODK3–ODY1. The comparison among the three explant types revealed that roots from petiole callus were the best source for increased biomass growth, RA accumulation and YIs. Roots from root callus were the least appropriate explant type exhibiting the lowest YIs, while roots from leaf callus gave intermediate results compared to the other two explant types (Table 4, Figure 3a–c).

## 3. Discussion

All adventitious roots produced contained the same metabolite, RA, in high amounts in the descending order of the initial explant petiole callus > leaf callus > root callus. Comparison with the initial plant tissues shows marked qualitative and quantitative differences. The aerial parts of *Origanum dictamnus* are a rich source of a variety of secondary metabolites, comprising RA; other depsides, like salvianolic acid P; and flavonoids. Previous chemical analyzes in our laboratory led to the characterization of more than forty secondary metabolites from extracts of different polarities [36]. Roots, on the other hand, have a simpler metabolic profile, as shown by preliminary chemical analyzes, with RA as the main representative and few phenolic acids (Appendix A). Roots, however, are difficult to harvest and have a lower yield of RA production; therefore, the aerial parts of the plant are a better starting material for further exploitation. Nevertheless, isolation of RA from a mature plant is a laborious task requiring many isolation steps, organic solvents and high cost. Furthermore, as with most secondary metabolites, climatic variations deeply affect metabolite production in plants. Green factories based on cell and organ cultures are emerging as an alternative bio-sustainable and eco-friendly source of high-value bioactive plant secondary metabolites, including RA [37]. According to Bais et al. [38], biotechnological production of RA in plant cell cultures is favored because the compound belongs to the so-called preformed small molecules, that is, metabolites that are persistently biosynthesized. Indeed, analyses of the roots of *O. dictamnus* revealed that the compound already existed (Appendix A).

The use of a plant growth regulator (PGR) [39], the chemical composition of the basal nutrient culture medium [40] and the explant type [41] are factors that influence the plant regeneration potential. In the present study, callus and root regeneration at a 100% rate were noticed at the same time by leaf, petiole and root *O. dictamnus* explants cultured in ODK3 and ODK4 media.

The effect of PGRs in culture media, when altered, has strong effects on the growth and the accumulation of different groups of secondary metabolites [42,43], depending on plant species [44,45]. In the studied *O. dictamnus*, between the two liquid culture media tested (ODY1, ODY2), the ODY1 (35 μM IBA + 2.5 μM kinetin, auxin/cytokinin ratio of 14:1 *v*/*v*) gave significantly higher biomass yields (FW, DW), % RA production and yield index for all three explant types (roots derived from leaf, petiole and root cultures) as compared to ODY2 (5 μM NAA + 2.5 μM kinetin, auxin/cytokinin ratio of 2:1 *v*/*v*) after 240 days of the total in vitro culture. According to Mehrotra et al. [46], continuous shaking on liquid media can generate enough oxygen supply until it finally weighs with fast and abundant growth, expediting the uniform distribution of nutrients for the entire explants, which ends up with the best root growth. A closer relationship between liquid culture media and tissue is achieved, leading to enhanced root growth due to greater impulse, faster uptake and immediate utilization of nutrients and hormones from plant tissues [47]. Both ODY1 and ODY2 liquid culture media were supplemented with the same type (kinetin) and concentration (2.5 μM) of cytokinin but differed in the type and concentration of the auxin (35 μM IBA in ODY1 vs. 5 μM NAA in ODY2); thus the concentration of auxin source in ODY1 was seven-fold higher than that in ODY2. Different findings have been reported by Korkor et al. [48], who found that the highest fresh and dry biomass was obtained from adventitious root cultures of *Origanum majorana* L. on a liquid MS medium containing 2.7 μM NAA as compared to IBA (0.5–2.5 μΜ) and lower NAA concentration (0.5 μM). According to Hartmann et al. [49], IBA is the strongest, most stable and less toxic form of auxin that is widely used as a root booster hormone for most species. IBA, compared to other types of auxin (NAA, IAA), is a hormone that is very suitable for the adventitious root culture of several medicinal plants such as 35 μM IBA in *Centella asiatica* [50], 25 μM IBA in *Labisia pumila* [51] and 2.5 μM in *Costus igneus* [52], pointing out that the dosage of auxin is one of a key factor in the induction of adventitious rooting in liquid culture media. The differences in fresh and dry biomass yields among the three different root-derived explant types used in this study can be attributed to the difference in the biomass growth rate potential of different explants, the phytochemical composition of the explants, the concentration/combination of endogenous PGRs and, above all, the genotype used [53]. There are reports pointing out that differences in the regenerative abilities of various parts of explants may be ascribed to the differences in endogenous hormones and nutrients [54], and these variations in the concentration of endogenous hormones accordingly influence the demand for exogenous hormones in tissue culture.

RA biosynthesis under in vitro conditions was also confirmed for several species and diverse in vitro culture systems such as callus cultures, suspension cultures, shoot cultures and hairy roots cultures [55]. According to Benedec et al. [56], 12.4 mg/g dry extract RA was quantified by an HPLC-MS method in *Origanum vulgare* L. while Kintzios et al. [57] reported increased production of RA in *Ocinum basilicum* in vitro cultures grown on media supplemented with NAA in combination with other PGRs. The comparison among the three root explant types, the two solid and the two liquid media compositions, as well as the different incubation periods in the solid media of the *O. dictamnus* under study, showed that the ODY1 liquid medium and the 240 days longest total culture duration is the optimum transition combination treatment promoting best biomass, % RA production and yield index, regardless explant type. In in vitro culture systems, the determination of the accumulated RA does not seem to be linked to the degree of culture differentiation but is highly dependent on plant species, cell line and age [58].

Based on the findings of Bauer et al. [59], in some cases, an inversely proportional relationship might be noticed between the growth of in vitro plant tissue cultures and RA accumulation, segregating the biomass growth from bioactive compound production. PGRs may prompt the growth of the culture and increase plant biomass but without simultaneously raising the production of secondary metabolites [60]. Regarding *O. dictamnus* adventitious in vitro root cultures under investigation, the combination of ODK4–ODY1 resulted in higher % RA content, biomass production and yield index for roots derived from leaf calluses and root calluses, whereas for petiole callus roots, the ODK3–ODY1 transition medium treatment was more beneficial, all under the longest culture duration period of 240 days.

In the case of leaf callus and root callus root cultures for 240 days divided into 200 days in solid ODK3 and another 40 days to liquid ODY2, a significant drop was noticed in % RA but without a simultaneous reduction in biomass yields as compared to shorter incubation periods, being 80 and 160 days. Biomass growth of plant tissues or cells may continue up to a certain point, regardless of cell division cessation and oxygen scarcity in the culture medium, provoking stress to the cells [61], thus influencing the cell growth rate, the maximum cell mass to a certain physical space and the distribution size of cell aggregates.

## 4. Materials and Methods

### 4.1. In vitro Adventitious Root Culture Establishment

Seeds were collected from mother plants maintained at Balkan Botanic Garden of Kroussia, Institute of Plant Breeding and Genetic Resources, Hellenic Agricultural Organization (ELGO)—DIMITRA, with the unique IPEN (International Plant Exchange Network) accession number GR-1-BBGK-03,2108 and in vitro culture of *O. dictamnus* were established according to the protocol of Sarropoulou et al. [62]. From the stock cultures, three different types of explants were used: (1) leaves, (2) petioles and (3) roots. Two different culture media differing in plant growth regulators (PGRs) concentration ratio [kinetin (Duchefa Biochemie B.V., Haarlem, The Netherlands)/α-naphthalene-acetic acid (NAA) (Sigma-Aldrich, St. Louis, MO, USA, suitable for plant cell culture, BioReagent, ≥95%, crystalline, Merck KGaA, Darmstadt, Germany] were tested ODK3 (5 μM NAA + 5 μM kinetin), and ODK4 (5 μM NAA + 0.5 μM kinetin). Basic medium used was the MS (Duchefa Biochemie B.V., Haarlem, The Netherlands) [63] supplemented with 30 g L^−1^ sucrose (Duchefa Biochemie B.V., Haarlem, The Netherlands) and 6 g L^−1^ Plant Agar (Duchefa Biochemie B.V., Haarlem, The Netherlands), pH 5.8. All explants were cultured in Magenta vessels containing 35 mL of medium, and cultures were kept in a growth chamber, T 22 ± 1 °C, in continuous dark. After 60 days, callus formation (%) (number of explants with callus formation/total number of initial explants × 100%) and shoot and root regeneration (%) percentages were recorded.

### 4.2. In vitro Adventitious Root Culture

Segments of adventitious roots derived from leaf callus, petiole callus and root callus cultures in ODK3 and ODK4 agar solidified media were used as explants for further culture into 250 mL Erlenmeyer flasks containing 100 mL of liquid media on a rotary shaker (120 rpm, 24 h dark, 22 ± 1°C). Two different liquid media were tested ODY1 with 35 μM IBA (Duchefa Biochemie B.V., Haarlem, The Netherlands) + 2.5 μM kinetin (Duchefa Biochemie B.V., Haarlem, The Netherlands) and ODY2 with 5 μM NAA (Sigma-Aldrich, suitable for plant cell culture, BioReagent, ≥95%, crystalline, Merck KGaA, Darmstadt, Germany) + 0.5 μM kinetin (Duchefa Biochemie B.V., Haarlem, The Netherlands). Basal liquid medium used was the MS (Duchefa Biochemie B.V., Haarlem, The Netherlands) enriched with 30 g L^−1^ sucrose (Duchefa Biochemie B.V., Haarlem, The Netherlands), pH 5.8. Each treatment consisted of three flasks with 25 explants; thus, n = 75 explants per treatment. The initial fresh weight of the 25 explants/flask was 1 g.

Experiments lasted for three different periods: 40, 160 and 200 days in agar solidified media mentioned above, followed by 40 days in liquid media, thus three in vitro total culture periods of 80 (40 + 40), 160 (120 + 40) and 240 (200 + 40) days, respectively.

Biomass, RA content (%) and yield index parameters were recorded for each type of explant (roots from leaf callus, roots from petiole callus, roots from root callus), the initial agar solidified medium (ODK3, ODK4), the composition of the liquid culture medium of the newly developed adventitious roots (ODY1, ODY2) and the period of the culture.

The following parameters were recorded (i) root fresh biomass growth rate (=final FW/initial FW), which is equivalent to fresh weight (FW) (g), (ii) dry weight (DW) (g), (iii) FW/DW ratio, (iv) relative amount of RA expressed as % *w*/*w* (=mg RA/100 mg DW × 100%), (v) mean yield index (YI) (mg/100 mL liquid medium/250 mL flask) and (vi) yield index (YI) per explant (mg/100 mL liquid medium/250 mL flask) = mean yield index/25 explants per flask. Similarly, the accumulated content (mg RA/100 mg DW) is a relative measurement of RA; thus, RA relative amount (% *w*/*w*) = RA accumulated content (mg RA/100 mg DW) × 100. Therefore, the yield index (mg RA/100 mL liquid medium/250 mL flask) is the absolute amount of RA (mg) per flask and calculated by the following equations: DW (mg) × RA relative amount (% *w*/*w*)/100 or DW (mg) × RA accumulated content (mg RA/100 mg DW). The DW (g) and FW/DW ratio were calculated after drying biomass in an air dryer at 40 °C for 48 h.

### 4.3. Sample Preparation for HPLC Analysis

100 mg of the lyophilized adventitious roots of *Origanum dictamnus* were accurately weighted and ultrasonicated with 10 mL methanol 100% for 30 min. The extraction was carried out once. The samples were filtered through paper filter, and the filtrates were adjusted to 100.0 mL in a volumetric flask using methanol 100%. The solutions were filtered through Nylon filters (0.45 µm pore size) and immediately injected.

### 4.4. Chemicals for the HPLC Analysis

All solvents used for the analysis were of HPLC grade and were purchased from Sigma-Aldrich. Water was purified by a Milli-Qplus system from Millipore. Nylon filters (0.45 µm pore size) were manufactured from Membrane Solutions L.L.C. (Auburn, Washington D.C., USA). Rosmarinic acid (purity 97%) was manufactured from Alfa Aesar GmbH & Co KG (subsidiary/affiliated company of Thermo Fisher Scientific, Erlenbachweg, Kandel, Germany). A stock solution of rosmarinic acid (1.26 mg/mL) was prepared in DMSO 100% and kept at −20°C. A series of dilutions (5–50 times) were prepared in methanol 100% and immediately subjected to analysis.

### 4.5. HPLC-PDA-MS Analysis Instrumentation and Quantitative Determination of Rosmarinic Acid

Analysis was carried out using an HPLC-PDA-MS Thermo Finnigan system (LC Pump Plus, Autosampler, Surveyor PDA Plus Detector) interfaced with an ESI MSQ Plus (manufacturer: Thermo Finnigan L.L.C., subsidiary/affiliated company of Thermo Fisher Scientific, Massachusetts, USA) and equipped with Xcalibur software. The mass spectrometer operated in the negative ionization mode. Scan spectra were from *m*/*z* 100 to 800. Gas temperature was at 350 °C. Nitrogen flow rate was 10 L/min, and capillary voltage was 3000 V. The cone voltage was 80 V. The column was an SB-Aq (Agilent) RP-C18 column (150 mm × 3 mm) with a particle size of 5 µm maintained at 30 °C. The eluents were H_2_O at pH 2.8 by formic acid (0.05% *v*/*v*) (A) and acetonitrile (B) and with a flow rate of 0.4 mL/min. Samples were analyzed using a gradient program as follows: 0–5 min, 85%A; 5–15 min, 85–78%A; 15–20 78%A; 20–22 min, 78–75%A; 22–27 min, 75%A; 27–37 min, 75–60%A; 37–44 min, 60%A; 44–48 min, 60–85%A; 48–53 min, 85%A. The injected volume of the samples was 5 μL of solution. The UV–vis spectra were recorded between 220 and 600 nm, and the chromatographic profiles were registered at 330 nm. For the quantitative determination of rosmarinic acid, the method of external standard was used. The regression curve was obtained by five different concentration levels and measuring each point in triplicate. Measurements were performed at 330 nm, which is the maximum absorbance of rosmarinic acid.

### 4.6. Sample Preparation for NMR Analysis

An aliquot of various samples obtained by the procedure described in previous Section 4.5. was evaporated to dryness and redissolved in deuterated methanol (CD3OD). On an Agilent DD2 500 (500.1 MHz for 1H-NMR) spectrometer, at 295 K in CD3OD, 1H NMR was measured.

### 4.7. Statistical Analysis

The experimental layout was completely randomized. The means were subjected to analysis of variance (ANOVA) and compared using the Duncan multiple-range test (*p* < 0.05) using the statistical program SPSS 17.0 (SPSS Inc., Illinois, New York, NY, USA).

In the case of the in vitro cultures in agar solidified media, the indirect regeneration experiment included two solid culture media (ODK3, ODK4), three types of explants (leaves, petioles, roots) with 25 explants per treatment and per explant type, 5 groups Magenta vessels × 5 explants per vessel, i.e., 25 leaves, 25 petioles and 25 roots, in total; therefore the experimental layout was a 2 × 3 factorial consisted of six treatments (3 explant types and 2 solid culture media).

In the case of the in vitro liquid leaf, petiole and root callus-derived adventitious root cultures for RA production, the experimental design was a 2 × 2 × 3 factorial one including two solid media (ODK3, ODK4), two liquid media (ODY1, ODY2) and three incubation periods in agar solidified media (40, 120, 200 days); thus, 12 treatments were tested. For all three induced adventitious roots used derived from three different callus types (leaf callus, petiole callus, root callus), the main effect of factors [initial agar solidified medium (A), liquid medium composition (B), culture duration in agar medium (C)] and their interactions (AxB, AxC, BxC, AxBxC) were determined by General Linear Model/three-way ANOVA (Table 1, Table 2 and Table 3).

In the following stage, having found that between the different liquid media and different incubation periods in agar solidified media, the liquid ODY1 and the longest culture duration of 200 days exhibited the best results related to biomass growth and RA production for induced adventitious roots derived from all three different leaf, petiole and root callus explant types, the most promising treatments were selected and subjected to a statistical analysis for means comparison. In this framework, the experimental design was a 3 × 2 factorial one consisting of three explant types (roots derived from leaf, petiole and root callus) cultured on liquid ODY1 medium (40 days) and two agar solidified media (ODK3, ODK4) (200 days) under the same total incubation period of 240 days. The main effect of factors [explant type (A), solid culture medium composition (B)] and their interaction (AxB) was determined by General Linear Model/three-way ANOVA) (Table 4).

As concerns the liquid in vitro cultures, in all Tables, means (n = 3 or 3 flasks × 25 explants/flask/treatment) ± standard error (S.E.) with the same letter in a column are not statistically significantly different from each other according to Duncan’s multiple range test at *p* ≤ 0.05. ns *p* > 0.05; * *p* ≤ 0.05; ** *p* ≤ 0.01; *** *p* ≤ 0.001.

## 5. Conclusions

The type of initial explant can influence the production of adventitious roots as well as the production of secondary metabolites in vitro in liquid culture media. The production of rosmarinic acid in *Origanum dictamnus* could be obtained in vitro by culturing adventitious roots. Further optimization of the methodology for scaling up production in bioreactors is underway.

## Figures and Tables

**Figure 1 plants-12-00299-f001:**
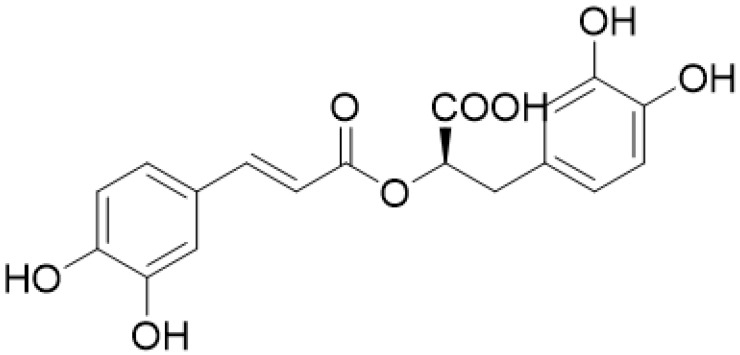
Structure of rosmarinic acid.

**Figure 2 plants-12-00299-f002:**
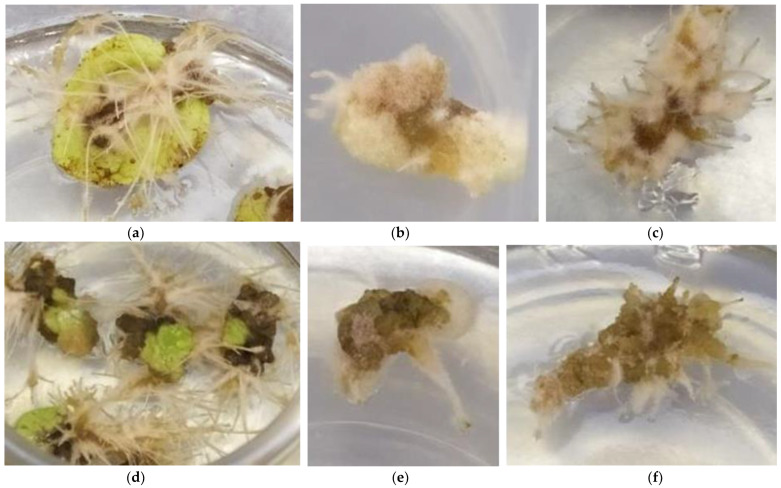
In vitro culture of three explant types (leaves, petioles, roots) of *Origanum dictamnus* L. in different media after two months: (**a**) root regeneration from leaves in ODK3, (**b**) callus and root formation from petiole explants in ODK3, (**c**) callus and new roots induction on root explants in ODK3 and (**d**–**f**) callus and root induction from leaf, petiole and root explants in ODK4, respectively.

**Figure 3 plants-12-00299-f003:**
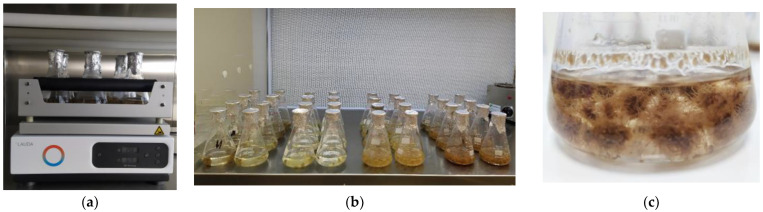
In vitro culture of *Origanum dictamnus* L. roots in liquid media: (**a**) flasks onto continuous rotary shaker agitated at 120 rpm, (**b**) all treatments into flasks onto laminar hood flow and (**c**) root biomass growth and formation of new adventitious roots over primary ones after 40 days of culture.

**Table 1 plants-12-00299-t001:** Effect of media composition and combinations, and three incubation periods on biomass growth (FW, DW, FW/DW ratio), and RA production expressed as % *w*/*w*, mean yield index and yield index per explant adventitious root in vitro cultures of *Origanum dictamnus* L. derived from leaf callus.

Explant Type	Solid Medium	Liquid Medium	In Vitro Culture Duration (in Days)			F.W. (g) (=Fresh Biomass Growth Rate)	D.W. (g)	F.W./D.W. Ratio	RA (% *w*/*w*)	Mean Yield Index *	Yield Index/Explant **
			Solid Medium	Liquid Medium	Total						
Leaf callus derived roots	ODK3	ODY1	40	40	80	2.8 ± 1.0 e	0.4 ± 0.1 e	7.4 ± 0.0 ef	1.5 ± 0.5 de	5.8 ± 0.9 e	0.23 ± 0.04 d
			120	40	160	2.7 ± 0.1 e	0.2 ± 0.0 e	6.5 ± 0.2 h	2.2 ± 0.0 cd	4.6 ± 0.0 f	0.18 ± 0.00 d
			200	40	240	15.2 ± 0.0 b	1.5 ± 0.0 b	10.1 ± 0.0 b	5.8 ± 0.6 a	86.7 ± 8.7 a	3.47 ± 0.35 a
	ODK4	ODY1	40	40	80	2.1 ± 0.5 e	0.3 ± 0.1 e	7.3 ± 0.0 fg	0.5 ± 0.0 e	1.5 ± 0.3 g	0.06 ± 0.01 e
			120	40	160	3.8 ± 0.5 de	0.5 ± 0.1 e	7.8 ± 0.2 e	4.1 ± 0.0 b	19.9 ± 2.5 cd	0.80 ± 0.10 bcd
			200	40	240	19.0 ± 0.2 a	1.8 ± 0.0 a	10.3 ± 0.0 b	5.1 ± 0.6 a	93.7 ± 12.2 a	3.75 ± 0.49 a
	ODK3	ODY2	40	40	80	4.4 ± 2.4 de	0.6 ± 0.2 e	11.3 ± 0.0 a	3.8 ± 0.1 b	22.3 ± 7.9 c	0.89 ± 0.32 bcd
			120	40	160	4.9 ± 0.3 de	0.5 ± 0.0 e	8.9 ± 0.0 c	5.2 ± 0.1 a	28.2 ± 1.9 c	1.13 ± 0.07 bc
			200	40	240	6.6 ± 0.0 cd	0.7 ± 0.0 d	8.9 ± 0.0 c	1.7 ± 0.4 d	12.3 ± 2.6 d	0.49 ± 0.11 cd
	ODK4	ODY2	40	40	80	2.5 ± 1.4 e	0.4 ± 0.2 e	6.3 ± 0.0 h	3.1 ± 0.1 bc	12.1 ± 5.8 de	0.48 ± 0.31 cd
			120	40	160	3.1 ± 0.7 e	0.4 ± 0.1 e	7.2 ± 0.2 g	3.9 ± 0.5 b	16.9 ± 5.2 cd	0.68 ± 0.21 cd
			200	40	240	8.3 ± 0.0 c	1.0 ± 0.0 c	8.5 ± 0.0 d	3.7 ± 0.4 b	36.4 ± 4.3 b	1.46 ± 0.17 b
Three-way ANOVA/General Linear Model											
Solid culture medium composition [ODK3 vs. ODK4] (A)						0.529 ns	0.256 ns	0.000 ***	0.792 ns	0.165 ns	0.165 ns
Liquid culture medium composition [ODY1 vs. ODY2] (B)						0.000 ***	0.001 **	0.000 ***	0.091 ns	0.000 ***	0.000 ***
In vitro culture duration in solid medium [40, 120, 200 days] (C)						0.000 ***	0.000 ***	0.000 ***	0.000 ***	0.000 ***	0.000 ***
(A) × (B)						0.074 ns	0.642 ns	0.000 ***	0.904 ns	0.931 ns	0.931 ns
(A) × (C)						0.012 *	0.069 ns	0.000 ***	0.014 *	0.070 ns	0.070 ns
(B) × (C)						0.000 ***	0.000 ***	0.000 ***	0.000 ***	0.000 ***	0.000 ***
(A) × (B) × (C)						0.821 ns	0.641 ns	0.000 ***	0.000 ***	0.078 ns	0.078 ns

FW: fresh weight, Fresh biomass growth rate = final FW/initial FW, DW: dry weight, RA: rosmarinic acid, * Mean yield index (mg/100 mL liquid medium/250 mL flask): DW (mg) × RA % (*w*/*w*)/100 or DW (mg) × mg RA/100 mg DW, ** Yield index/explant (mg/100 mL liquid medium/250 mL flask): Mean yield index/25 explants per flask per treatment. FW is equivalent to fresh biomass growth rate as the initial FW was 1 g. Means (n = 3 or 3 flasks × 25 explants/flask/treatment) ± standard error (S.E.) with the same letter in a column are not statistically significantly different from each other according to Duncan’s multiple range test at *p* ≤ 0.05. ns *p* > 0.05; * *p* ≤ 0.05; ** *p* ≤ 0.01; *** *p* ≤ 0.001.

**Table 2 plants-12-00299-t002:** Effect of media composition and combinations, and two incubation periods on biomass growth (FW, DW, FW/DW ratio), and RA production expressed as % *w*/*w*, mean yield index and yield index per explant adventitious root in vitro cultures of *Origanum dictamnus* L. derived from petiole callus.

Explant Type	Solid Medium	Liquid Medium	In Vitro Culture Duration (in Days)			F.W. (g) (=Fresh Biomass Growth Rate)	D.W. (g)	F.W./D.W. Ratio	RA (% *w*/*w*)	Mean Yield Index *	Yield Index/Explant **
			Solid Medium	Liquid Medium	Total						
Petiole callus derived roots	ODK3	ODY1	120	40	160	3.1 ± 0.0 d	0.4 ± 0.0 d	8.4 ± 0.2 b	1.8 ± 0.0 d	6.8 ± 0.1 f	0.27 ± 0.01 e
			200	40	240	18.8 ± 1.0 a	1.8 ± 0.1 a	10.7 ± 0.0 a	6.6 ± 0.0 a	115.3 ± 5.8 a	4.61 ± 0.23 a
	ODK4	ODY1	120	40	160	4.2 ± 0.7 d	0.5 ± 0.1 d	7.8 ± 0.1 bc	3.5 ± 0.0 c	18.8 ± 3.1 e	0.75 ± 0.12 d
			200	40	240	16.6 ± 0.4 b	1.6 ± 0.2 a	10.4 ± 0.9 a	5.6 ± 0.4 b	89.9 ± 15.3 b	3.60 ± 0.61 b
	ODK3	ODY2	120	40	160	3.8 ± 0.6 d	0.5 ± 0.1 d	8.3 ± 0.2 bc	3.6 ± 0.0 c	16.6 ± 3.3 e	0.66 ± 0.13 d
			200	40	240	7.2 ± 0.1 c	0.8 ± 0.0 c	8.6 ± 0.0 b	3.7 ± 0.1 c	31.1 ± 1.2 d	1.25 ± 0.05 c
	ODK4	ODY2	120	40	160	3.3 ± 0.2 d	0.5 ± 0.0 d	7.2 ± 0.2 c	1.7 ± 0.0 d	7.8 ± 0.7 f	0.31 ± 0.03 e
			200	40	240	8.8 ± 0.5 c	1.1 ± 0.1 b	8.0 ± 0.0 bc	5.6 ± 0.3 b	61.1 ± 6.7 c	2.44 ± 0.70 b
Three-way ANOVA/General Linear Model											
Solid culture medium composition [ODK3 vs ODK4] (A)						0.979 ns	0.233 ns	0.023 *	0.225 ns	0.703 ns	0.000 ***
Liquid culture medium composition [ODY1 vs. ODY2] (B)						0.000 ***	0.000 ***	0.000 ***	0.000 ***	0.000 ***	0.490 ns
In vitro culture duration in solid medium [120, 200 days] (C)						0.000 ***	0.000 ***	0.000 ***	0.000 ***	0.000 ***	0.000 ***
(A) × (B)						0.143 ns	0.287 ns	0.436 ns	0.104 ns	0.066 ns	0.000 ***
(A) × (C)						0.417 ns	0.794 ns	0.362 ns	0.029 *	0.879 ns	0.000 ***
(B) × (C)						0.000 ***	0.000 ***	0.001 **	0.000 ***	0.000 ***	0.433 ns
(A) × (B) × (C)						0.003 **	0.028 *	0.935 ns	0.000 ***	0.001 **	0.000 ***

FW: fresh weight, Fresh biomass growth rate = final FW/initial FW, DW: dry weight, RA: rosmarinic acid, * Mean yield index (mg/100 mL liquid medium/250 mL flask): DW (mg) × RA % (*w*/*w*)/100 or DW (mg) × mg RA/100 mg DW, ** Yield index/explant (mg/100 mL liquid medium/250 mL flask): Mean yield index/25 explants per flask per treatment. FW is equivalent to fresh biomass growth rate as the initial FW was 1 g. Means (n = 3 or 3 flasks × 25 explants/flask/treatment) ± standard error (S.E.) with the same letter in a column are not statistically significantly different from each other according to Duncan’s multiple range test at *p* ≤ 0.05. ns *p* > 0.05; * *p* ≤ 0.05; ** *p* ≤ 0.01; *** *p* ≤ 0.001.

**Table 3 plants-12-00299-t003:** Effect of media composition and combinations, and three incubation periods on biomass growth (FW, DW, FW/DW ratio), and RA production expressed as % *w*/*w*, mean yield index and yield index per explant adventitious root in vitro cultures of *Origanum dictamnus* L. derived from root callus.

Explant Type	Solid Medium	Liquid Medium	In Vitro Culture Duration (in Days)			F.W. (g) (=Fresh Biomass Growth Rate)	D.W. (g)	F.W./D.W. Ratio	RA (% *w*/*w*)	Mean Yield Index *	Yield Index/ Explant **
			Solid Medium	Liquid Medium	Total						
Root callus derived roots	ODK3	ODY1	40	40	80	4.1 ± 0.0 d	0.5 ± 0.0 e	9.0 ± 0.5 bcd	1.1 ± 0.0 i	5.0 ± 0.4 e	0.20 ± 0.01 e
			120	40	160	4.3 ± 0.3 d	0.4 ± 0.0 ef	12.1 ± 1.6 a	2.6 ± 0.0 f	9.3 ± 0.9 d	0.37 ± 0.03 d
			200	40	240	16.2 ± 0.6 a	1.7 ± 0.1 a	9.8 ± 0.0 bc	2.9 ± 0.0 d	47.4 ± 2.2 a	1.90 ± 0.09 a
	ODK4	ODY1	40	40	80	1.1 ± 0.0 g	0.1 ± 0.0 g	7.8 ± 0.6 cde	1.1 ± 0.0 i	1.5 ± 0.4 f	0.06 ± 0.02 f
			120	40	160	2.4 ± 0.3 e	0.4 ± 0.0 ef	6.8 ± 0.1 e	1.6 ± 0.0 h	5.5 ± 0.6 e	0.22 ± 0.03 e
			200	40	240	11.6 ± 0.4 b	1.1 ± 0.0 b	10.7 ± 0.3 ab	4.7 ± 0.0 a	51.4 ± 1.4 a	2.06 ± 0.01 a
	ODK3	ODY2	40	40	80	1.1 ± 0.1 g	0.1 ± 0.0 g	8.7 ± 0.6 bcd	1.1 ± 0.0 i	1.4 ± 0.4 f	0.06 ± 0.01 g
			120	40	160	3.0 ± 0.6 e	0.4 ± 0.1 ef	7.6 ± 0.1 de	3.2 ± 0.0 c	12.4 ± 1.8 c	0.50 ± 0.07 c
			200	40	240	6.6 ± 0.3 c	0.8 ± 0.0 c	8.2 ± 0.3 cde	2.7 ± 0.0 e	21.9 ± 1.8 b	0.88 ± 0.07 b
	ODK4	ODY2	40	40	80	1.5 ± 0.0 fg	0.2 ± 0.0 g	8.4 ± 0.3 cde	0.8 ± 0.0 j	1.5 ± 0.0 f	0.06 ± 0.00 f
			120	40	160	2.4 ± 0.0 ef	0.3 ± 0.0 f	7.2 ± 0.3 de	2.0 ± 0.0 g	6.6 ± 0.7 e	0.26 ± 0.03 e
			200	40	240	5.8 ± 0.1 c	0.6 ± 0.0 d	9.5 ± 0.0 bc	3.3 ± 0.0 b	20.4 ± 0.2 b	0.82 ± 0.01 b
Three-way ANOVA/General Linear Model											
Solid culture medium composition [ODK3 vs. ODK4] (A)						0.000 ***	0.000 ***	0.013 *	0.608 ns	0.003 **	0.003 **
Liquid culture medium composition [ODY1 vs. ODY2] (B)						0.000 ***	0.000 ***	0.002 **	0.000 ***	0.000 ***	0.000 ***
In vitro culture duration in solid medium [40, 120, 200 days] (C)						0.000 ***	0.000 ***	0.023 *	0.000 ***	0.000 ***	0.000 ***
(A) × (B)						0.000 ***	0.000 ***	0.004 **	0.000 ***	0.321 ns	0.321 ns
(A) × (C)						0.004 **	0.000 ***	0.000 ***	0.000 ***	0.004 **	0.004 **
(B) × (C)						0.000 ***	0.000 ***	0.022 *	0.000 ***	0.000 ***	0.000 ***
(A) × (B) × (C)						0.015 *	0.000 ***	0.011 *	0.000 ***	0.019 *	0.019 *

FW: fresh weight, Fresh biomass growth rate = final FW/initial FW, DW: dry weight, RA: rosmarinic acid, * Mean yield index (mg/100 mL liquid medium/250 mL flask): DW (mg) × RA % (*w*/*w*)/100 or DW (mg) × mg RA/100 mg DW, ** Yield index/explant (mg/100 mL liquid medium/250 mL flask): Mean yield index/25 explants per flask per treatment. FW is equivalent to fresh biomass growth rate as the initial FW was 1 g. Means (n = 3 or 3 flasks × 25 explants/flask/treatment) ± standard error (S.E.) with the same letter in a column are not statistically significantly different from each other according to Duncan’s multiple range test at *p* ≤ 0.05. ns *p* > 0.05; * *p* ≤ 0.05; ** *p* ≤ 0.01; *** *p* ≤ 0.001.

**Table 4 plants-12-00299-t004:** Effect of three explant types (leaf, petiole and root callus derived roots) and initial agar solidified media composition under 240 days of total incubation period on biomass and RA production in adventitious root in vitro cultures of *Origanum dictamnus* L.

Explant Type	Solid Medium	Liquid Medium	In Vitro Culture Duration (in Days)			F.W. (g) (=Fresh Biomass Growth Rate)	D.W. (g)	F.W./D.W. Ratio	RA (% *w*/*w*)	Mean Yield Index *	Yield Index/Explant **
			Solid Medium	Liquid Medium	Total						
Leaf callus derived roots	ODK3	ODY1	200	40	240	15.2 ± 0.0 b	1.5 ± 0.0 b	10.1 ± 0.0 a	5.8 ± 0.6 ab	86.7 ± 8.7 b	3.47 ± 0.35 b
	ODK4		200	40	240	19.0 ± 0.2 a	1.8 ± 0.0 a	10.3 ± 0.0 a	5.1 ± 0.6 b	93.7 ± 12.2 b	3.75 ± 0.49 b
Petiole callus derived roots	ODK3	ODY1	200	40	240	18.8 ± 1.0 a	1.8 ± 0.1 ab	10.7 ± 0.0 a	6.6 ± 0.0 a	115.3 ± 5.8 a	4.61 ± 0.23 a
	ODK4		200	40	240	16.6 ± 0.4 b	1.6 ± 0.2 ab	10.4 ± 1.0 a	5.6 ± 0.4 ab	89.9 ± 15.3 b	3.60 ± 0.61 b
Root callus derived roots	ODK3	ODY1	200	40	240	16.2 ± 0.6 b	1.7 ± 0.1 ab	9.8 ± 0.0 a	2.9 ± 0.0 c	47.4 ± 2.2 c	1.90 ± 0.09 c
	ODK4		200	40	240	11.6 ± 0.4 c	1.1 ±0.0 c	10.7 ± 0.3 a	4.7 ± 0.0 b	51.4 ± 0.4 c	2.06 ± 0.01 c
Three-way ANOVA/General Linear Model											
Explant type [Roots derived from leaf callus, petiole callus, root callus] (A)						0.000 ***	0.005 **	0.501 ns	0.000 ***	0.000 ***	0.000 ***
Solid culture medium composition [ODK3 vs. ODK4] (B)						0.031 *	0.081 ns	0.350 ns	0.792 ns	0.495 ns	0.495 ns
(A) × (B)						0.000 ***	0.001 **	0.412 ns	0.004 **	0.186 ns	0.186 ns

FW: fresh weight, Fresh biomass growth rate = final FW/initial FW, DW: dry weight, RA: rosmarinic acid, * Mean yield index (mg/100 mL liquid medium/250 mL flask): DW (mg) × RA % (*w*/*w*)/100 or DW (mg) × mg RA/100 mg DW, ** Yield index/explant (mg/100 mL liquid medium/250 mL flask): Mean yield index/25 explants per flask per treatment. FW is equivalent to fresh biomass growth rate as the initial FW was 1 g. Means (n = 3 or 3 flasks × 25 explants/flask/treatment) ± standard error (S.E.) with the same letter in a column are not statistically significantly different from each other according to Duncan’s multiple range test at *p* ≤ 0.05. ns *p* > 0.05; * *p* ≤ 0.05; ** *p* ≤ 0.01; *** *p* ≤ 0.001.

## Data Availability

Not applicable.

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
