# Peer review of "Rosmarinic Acid Production from Origanum dictamnus L. Root Liquid Cultures In Vitro"

_plants, 2023, doi:10.3390/plants12020299_

Round 1

Reviewer 1 Report

The manuscript is correctly written, the experimental design is adequate, and the methods were accurately described. Roots are good platforms for producing secondary metabolites, it would be interesting if in the future you compare the performance of these adventitious roots with hairy roots that usually reach higher final biomass and metabolite yields. 

Author Response

Journal: Plants (ISSN 2223-7747)

Manuscript ID: plants-2084836

Type: Article

Title: Rosmarinic acid production from Origanum dictamnus L. root liquid cultures in vitro

Authors: Virginia Saropoulou, Charikleia Paloukopoulou, Anastasia Karioti, Eleni Maloupa, Katerina Grigoriadou

Section: Plant Physiology and Metabolism

Special Issue: Production of Secondary Metabolites In Vitro

Dear Reviewer #1,

All the authors appreciate the time spent for further suggestions in an attempt to improve the content of this manuscript. All comments were taken into consideration and corrected accordingly. We are grateful to receive reviews that help us improve the quality of our manuscript. Following your comments we corrected the text adopting all the comments suggested. In the revised manuscript, all changes and new additions in text and throughout the manuscript are highlighted with yellow color. In addition, in this response letter, the authors gave responses below each comment in yellow highlighted text in italics. See below in detail.

Reviewer #1

Comments and Suggestions for Authors

The manuscript is correctly written, the experimental design is adequate, and the methods were accurately described. Roots are good platforms for producing secondary metabolites, it would be interesting if in the future you compare the performance of these adventitious roots with hairy roots that usually reach higher final biomass and metabolite yields. 

Authors’ response: We appreciate the encouraging comments of the reviewer #1 as it give us the support to continue this study further. Indeed, our new step is the comparison of the accumulated content and productivity or yield index of the in vitro adventitious root (ARs) cultures presented herein on the one hand with ARs after treatment with biotic (i.e. pectin) and abiotic elicitors (i.e. temperature, heavy metals, different sucrose concentrations, proline, jasmonates, salicylic acid, among others) and on the other hand with hairy roots after transformation with Agrobacterium rhizogenes. The study of ARs under different elicitation treatments is already underway while the hairy root cultures soon will be feasible.

Reviewer 2 Report

The manuscript entitled “Rosmarinic acid production from Origanum dictamnus L. root liquid cultures in vitro” describes establishment of in vitro adventitious root culture system for production of rosmarinic acid. O. dictamnus is an endemic plant species grown in the mountainsides and gorges of the island of Crete in Greece.

The suggestions to authors to be considered are listed below. 

Page 2 line 84

The aim of the study was the investigation of a suitable in vitro adventitious root culture system for the production of RA from the Greek endemic Origanum dictamnus

Should be changed to

The aim of the study was the establishmentof a suitable in vitro adventitious root culture system for the production of RA from the Greek endemic Origanum dictamnus

Page 3 line 100

Table 1 should be removed as it does not carry additional information and presents data without a difference. The results described in the text (lines 92-94) are sufficient.

In this section, the authors can provide more information, if available, on which day callus formation started, on which cultural medium calli formed in greater quantities depending on the type of explant. Also, whether the induced adventitious roots differ morphologically.

In Discussion

Page 10 line 1

All adventitious roots produced contained the same metabolite, RA in high amounts in the descending order of the initial explant petiole-callus < leaf-callus < root-callus. – This sentence is not clear. Please rewrite. According to the text high amount of RA accumulated in roots derived from petiole-callus.

Page 10 line 23

(Chimdessa 2020) delete and indicate the reference with number 

Author Response

Journal: Plants (ISSN 2223-7747)

Manuscript ID: plants-2084836

Type: Article

Title: Rosmarinic acid production from Origanum dictamnus L. root liquid cultures in vitro

Authors: Virginia Saropoulou, Charikleia Paloukopoulou, Anastasia Karioti, Eleni Maloupa, Katerina Grigoriadou

Section: Plant Physiology and Metabolism

Special Issue: Production of Secondary Metabolites In Vitro

Dear Reviewer #2,

All the authors appreciate the time spent for further suggestions in an attempt to improve the content of this manuscript. All comments were taken into consideration and corrected accordingly. We are grateful to receive reviews that help us improve the quality of our manuscript. Following your comments we corrected the text adopting all the comments suggested. In the revised manuscript, all changes and new additions in text and throughout the manuscript are highlighted with yellow color. In addition, in this response letter, the authors gave responses below each comment in yellow highlighted text in italics. See below in detail.

Reviewer #2

Comments and Suggestions for Authors

The manuscript entitled “Rosmarinic acid production from Origanum dictamnus L. root liquid cultures in vitro” describes establishment of in vitro adventitious root culture system for production of rosmarinic acid. O. dictamnus is an endemic plant species grown in the mountainsides and gorges of the island of Crete in Greece. The suggestions to authors to be considered are listed below. 

Page 2 line 84: “The aim of the study was the investigation of a suitable in vitro adventitious root culture system for the production of RA from the Greek endemic Origanum dictamnus” should be changed to “The aim of the study was the establishment of a suitable in vitro adventitious root culture system for the production of RA from the Greek endemic Origanum dictamnus

Authors’ response: We followed your suggestion by replacing the word “investigation” with “establishment”, see revised manuscript, page 2, line 84.

Page 3 line 100: Table 1 should be removed as it does not carry additional information and presents data without a difference. The results described in the text (lines 92-94) are sufficient.

Authors’ response: We removed Table 1 as it was considered redundant. See text in revised manuscript, page 3, lines 108-111.

In this section, the authors can provide more information, if available, on which day callus formation started, on which cultural medium calli formed in greater quantities depending on the type of explant. Also, whether the induced adventitious roots differ morphologically.

Authors’ response: We inserted new text, in the same paragraph providing detailed description to all the questions, queries and recommendations as kindly requested by the reviewer. See revised manuscript, Results section, sub-section 2.2. In vitro adventitious root culture establishment, page 3, lines 111-137.

In Discussion, Page 10 line 1: All adventitious roots produced contained the same metabolite, RA in high amounts in the descending order of the initial explant petiole-callus < leaf-callus < root-callus. – This sentence is not clear. Please rewrite. According to the text high amount of RA accumulated in roots derived from petiole-callus.

Authors’ response: Indeed, we had made a typo using a wrong mathematical symbol. The sentence was corrected as “…in the descending order of the initial explant petiole-callus > leaf-callus > root-callus”, See revised manuscript, Discussion, page 10, line 3.

Page 10 line 23: (Chimdessa 2020) delete and indicate the reference with number 

Authors’ response: Deleted, the specific reference is indicated with number in brackets i.e. [40]. See revised manuscript, page 10, line 23.

Reviewer 3 Report

The presented study aims to find the best way to culture Origanum dictamnus adventitious root cultures to produce rosmarinic acid.

I have a few comments regarding mainly method description and presentation of the results:

1. How was the rate of callus formation determined? % of explants forming a callus?

2. What is the difference between the yield index and absolute amount of rosmarinic acid?

3. Why was the LC analysis so long? Was it necessary for the resolution of the one analyte?

4. Do lines 176-182 really apply only to the root cultures?

5. Do the paragraphs 170-174 and 183-192 describe different experiments?

6. In my opinion, Table 1 is redundant, and the results presented therein can only be mentioned in the text.

7. Including the results of rosmarinic acid analysis into the paragraph 2.1 seems not ideally oredered to me.

8. Please include the results of NMR into the main text and discuss them.

9. Please include the TLC analysis into the main text (method description, results, discussion) or do not mention it (L 98).

10. Was the effect of the length of cultivation in the liquid medium also tested? (Why not?)

Author Response

Journal: Plants (ISSN 2223-7747)

Manuscript ID: plants-2084836

Type: Article

Title: Rosmarinic acid production from Origanum dictamnus L. root liquid cultures in vitro

Authors: Virginia Saropoulou, Charikleia Paloukopoulou, Anastasia Karioti, Eleni Maloupa, Katerina Grigoriadou

Section: Plant Physiology and Metabolism

Special Issue: Production of Secondary Metabolites In Vitro

Dear Reviewer #3,

All the authors appreciate the time spent for further suggestions in an attempt to improve the content of this manuscript. All comments were taken into consideration and corrected accordingly. We are grateful to receive reviews that help us improve the quality of our manuscript. Following your comments we corrected the text adopting all the comments suggested. In the revised manuscript, all changes and new additions in text and throughout the manuscript are highlighted with yellow color. In addition, in this response letter, the authors gave responses below each comment in yellow highlighted text in italics. See below in detail.

Reviewer #3

Comments and Suggestions for Authors

The presented study aims to find the best way to culture Origanum dictamnus adventitious root cultures to produce rosmarinic acid.

I have a few comments regarding mainly method description and presentation of the results:

  1. How was the rate of callus formation determined? % of explants forming a callus?

Authors response: Callus formation percentage (%) = number of explants with callus formation/ total number of initial explants x 100%. The above was also inserted into manuscript text. See revised manuscript, 4. M&M section, 4.1. In vitro adventitious root culture establishment subsection, page 12, lines 108-109.

  1. What is the difference between the yield index and absolute amount of rosmarinic acid?

Authors’ response: As it can be seen written in Materials and methods section, in the 4th paragraph of subsection 4.2. In vitro adventitious root culture, the absolute amount of rosmarinic acid (RA) is expressed as % w/w = mg RA/ 100 mg DW x 100% (measurement unit of dry weight – mg RA /100 mg DW) whereas the yield index is an parameter of productivity or in other words of total production which is expressed in a different measurement unit of dry weight/volume (mg/ 100 mL liquid medium/ 250 mL flask) combining total dry weight (mg DW) x absolute amount of rosmarinic acid (mg rosmarinic acid per mg dry weight). Therefore, the absolute amount of RA represents the accumulated content (mg RA/100 mg DW) (measurement unit: dry weight) while yield index or productivity or total production (mg RA/ 100 mL liquid medium/ 250 mL flask) is represented by the total root dry biomass production combined with the RA accumulated content (measurement unit: dry weight/ volume). In this framework, the last sentence of the above text was also added in the revised manuscript, see page 12, lines 136-140.

  1. Why was the LC analysis so long? Was it necessary for the resolution of the one analyte?

Authors’ response: The HPLC method was long in purpose. The same methodology is used also for the analysis of the species Origanum dictamnus, which of course is richer in metabolites. There two reasons for which we kept this method and did not shorten it:

  • At earlier elution times simpler metabolites, like caffeic acid and 3,4-dihydroxyphenylactic acid, which are the building blocks of rosmarinic acid appear. Furthermore, at later elution times other esters, derivatives of rosmarinic acid also eluted (methyl ester of rosmarinic acid, salvianolic acid P, derivatives of salvianolic acid P, etc). So, we kept the long elution time in order to see whether there were other metabolites present in the samples.
  • As it can be seen form the NMR there were also fatty acids present in the culture, so a washing period with high percentage of acetonitrile was included, together with re-equilibration period in order to wash the HPLC column for maintenance reasons.

  1. Do lines 176-182 really apply only to the root cultures?

Authors’ response: We revised the specific points in M&M section, subsection 4.7. Statistical analysis in such a way so as to be more clearly stated and easily comprehensible to the reader. See revised manuscript, page 13, lines 183-202. 

  1. Do the paragraphs 170-174 and 183-192 describe different experiments?

Authors’ response: The different paragraphs in this subsection 4.7. Statistical analysis describe not different experiments but subsequent stages of the same experiment. In specific, the former paragraph lines are related to the indirect experiment of the in vitro cultures of different initial explant types (leaves, petioles, roots) in agar solidified media with the aim of callus, shoot and root regeneration whereas the latter paragraph lines are referred to the following stage of the in vitro cultures of the induced adventitious roots, in liquid media for rosmarinic acid production, which were derived from leaf-, petiole- and root-callus after 60 days of culture in agar solidified media. Therefore, we revised specific points and sentences in this subsection for more clarity. See changes in the revised manuscript, page 13 (lines 183-202) & page 14 (lines 203-209).

  1. In my opinion, Table 1 is redundant, and the results presented therein can only be mentioned in the text.

Authors’ response: Deleted, Table 1 does not exist in the revised manuscript, thus we re-arranged the numbering order in the other Tables pre-existed in the initial submission manuscript, accordingly. (i.e. Table 2 renamed as Table 1, Table 3 to Table 2, Table 4 to Table 3, and Table 5 to Table 4.

  1. Including the results of rosmarinic acid analysis into the paragraph 2.1 seems not ideally oredered to me.

Authors’ response: We have transferred it to a separate paragraph and a separate subsection. See revised manuscript, Results section, subsection 2.1. Evaluation of rosmarinic acid production by HPLC-UV and NMR, page 3, lines 91-105. 

  1. Please include the results of NMR into the main text and discuss them.

Authors’ response: A short description of the NMR is introduced in the main text (see page 3, lines 91-105) and a more detailed in the supplementary material (see Figure S2).

  1. Please include the TLC analysis into the main text (method description, results, discussion) or do not

mention it (L 98).

Authors’ response: Since this is a very trivial method, we have not kept the TLC plate. So, we decided to eliminate it from the main text.

  1. Was the effect of the length of cultivation in the liquid medium also tested? (Why not?)

Authors’ response: The experimental results data presented herein are described for the first time for this plant species, no previous published work exists, and we consider that the amount of findings is more than enough for a manuscript to be published. The effect of different incubation periods of the induced adventitious roots in combination with different biotic and/or abiotic elicitors in order to obtain higher accumulated content and total productivity of rosmarinic acid than those presented herein is our future goal. Preliminary trials and experiments are at the moment underway and some others have been designed and will be accomplished soon.

Reviewer 4 Report

Dear authors

in my capacity as a reviewer and with the aim of improving this manuscript, I would like to make the following recommendations: sub-chapter 3 Discussion -line 23- insert the number between brackets of the referent and check the numbering in the list of referents. Sub-chapter 4 Materials and Methods please specify in this manuscript the companies and countries that sold the chemicals used for in vitro studies.

Author Response

Journal: Plants (ISSN 2223-7747)

Manuscript ID: plants-2084836

Type: Article

Title: Rosmarinic acid production from Origanum dictamnus L. root liquid cultures in vitro

Authors: Virginia Saropoulou, Charikleia Paloukopoulou, Anastasia Karioti, Eleni Maloupa, Katerina Grigoriadou

Section: Plant Physiology and Metabolism

Special Issue: Production of Secondary Metabolites In Vitro

Dear Reviewer #4,

All the authors appreciate the time spent for further suggestions in an attempt to improve the content of this manuscript. All comments were taken into consideration and corrected accordingly. We are grateful to receive reviews that help us improve the quality of our manuscript. Following your comments we corrected the text adopting all the comments suggested. In the revised manuscript, all changes and new additions in text and throughout the manuscript are highlighted with yellow color. In addition, in this response letter, the authors gave responses below each comment in yellow highlighted text in italics. See below in detail.

Reviewer #4

Comments and Suggestions for Authors

Dear authors, in my capacity as a reviewer and with the aim of improving this manuscript, I would like to make the following recommendations: sub-chapter 3 Discussion -line 23- insert the number between brackets of the referent and check the numbering in the list of referents. Sub-chapter 4 Materials and Methods please specify in this manuscript the companies and countries that sold the chemicals used for in vitro studies.

Authors’ response:

  • The citation (Chimdessa 2020) within parenthesis in the text was deleted and replaced in its turn by the reference with number in brackets i.e. [40]. The numbering in the reference list was also checked. See revised manuscript, page 10, line 23.
  • In M&M section, in subsections 4.1. and 4.2. related to the in vitro culture studies, the companies and countries of chemicals used in this study were specified and provided in parenthesis after the name of the chemical product. See revised manuscript, pages 12-13.

Round 2

Reviewer 3 Report

I would like to thank the authors for their responses. Regrettably, I must admit that I still do not get the method and purpose of calculating the mean yield index. You state that: “the absolute amount of rosmarinic acid (RA) is expressed as % w/w = mg RA/ 100 mg DW x 100% (measurement unit of dry weight – mg RA /100 mg DW)“. As it is expressed in percent, it is a relative, not an absolute amount. Similarly, “the accumulated content (mg RA/100 mg DW)“ is not absolute amount of RA but a relative measure. If, e.g. (Table 3), DW is 500 mg and relative RA amount is 1.1 %, the absolute amount of RA is 5.5 mg (in one flask). Further, you state that: “Mean yield index (mg/ 100 mL liquid medium/ 250 mL flask): total DW (mg) x mg RA/mg DW“. What is the difference between “total DW“ and “DW“? If total DW and DW is the same (e.g. 500 mg), then mean yield index is simply the absolute amount of RA (mg) per flask (e.g. 5.5 mg). This should be clarified for less bright readers (like me).

Author Response

Journal: Plants (ISSN 2223-7747)

Manuscript ID: plants-2084836

Type: Article

Title: Rosmarinic acid production from Origanum dictamnus L. root liquid cultures in vitro

Authors: Virginia Saropoulou, Charikleia Paloukopoulou, Anastasia Karioti, Eleni Maloupa, Katerina Grigoriadou

Section: Plant Physiology and Metabolism

Special Issue: Production of Secondary Metabolites In Vitro

Reviewer #3

Comments and Suggestions for Authors

I would like to thank the authors for their responses. Regrettably, I must admit that I still do not get the method and purpose of calculating the mean yield index. You state that: “the absolute amount of rosmarinic acid (RA) is expressed as % w/w = mg RA/ 100 mg DW x 100% (measurement unit of dry weight – mg RA /100 mg DW)“. As it is expressed in percent, it is a relative, not an absolute amount. Similarly, “the accumulated content (mg RA/100 mg DW)“ is not absolute amount of RA but a relative measure. If, e.g. (Table 3), DW is 500 mg and relative RA amount is 1.1 %, the absolute amount of RA is 5.5 mg (in one flask). Further, you state that: “Mean yield index (mg/ 100 mL liquid medium/ 250 mL flask): total DW (mg) x mg RA/mg DW“. What is the difference between “total DW“ and “DW“? If total DW and DW is the same (e.g. 500 mg), then mean yield index is simply the absolute amount of RA (mg) per flask (e.g. 5.5 mg). This should be clarified for less bright readers (like me).

Authors’ response: Dear Reviewer #3,

Thank you again for the extra comment, as it gave us the motivation to describe better the methodology used for the calculation of yield index and provide a more clear definition for the different terms used in this study (i.e. RA accumulated content, RA relative amount, yield index as absolute RA amount) and the existing relationship among them. All comments were taken into consideration and corrected accordingly. Indeed, there is no difference between “total DW” and “DW”, in fact it represents the final mean DW per flask which was obtained in the end of the culture period, thus in the revised manuscript, we deleted the word “total” before “DW” in both text and Tables (1-4) footnotes keeping only “DW”. In the revised manuscript, modifications and new additions in text and throughout the manuscript are highlighted with yellow color based on your recommendations as kindly indicated. See: 2nd revised manuscript, Results section in pages 6,7,8,9 in the footnotes of Tables 1,2,3,4 respectively, as well as in M&M section, subsection 4.2. In vitro adventitious root culture, page 12, lines 133-141.   
